# Scientometric Analysis of Research in Energy Efficiency and Citizen Science through Projects and Publications

**Daniela De Filippo** [1,2,*] , **María Luisa Lascurain** [1,2], **Andres Pandiella-Dominique** [1]
**and Elias Sanz-Casado** [1,2]

1   INAECU Research Institute for Higher Education and Science (UC3M-UAM), Calle Madrid 126, 28903 Getafe,
    Spain; mlascura@bib.uc3m.es (M.L.L.); apandiel@bib.uc3m.es (A.P.-D.); elias@bib.uc3m.es (E.S.-C.)
2   Department of Library and Information Sciences, Carlos III University of Madrid, Calle Madrid 126,
    28903 Getafe, Spain
*   Correspondence: dfilippo@bib.uc3m.es

**Abstract:** Energy efficiency is part of the commitment to environmental sustainability made by the organizations that promote and finance research and by the researchers that make this field their subject of study. Although there is growing interest in the subject, it is worth asking whether the research has been approached considering citizens' needs or citizens' participation. The main objective of this study is to analyse whether energy efficiency research has adopted a citizen science perspective. Using scientometric methods, the SCOPUS and CORDIS databases were consulted and a document search strategy was developed to gather information on publications and projects. The analysis revealed that, out of 265 projects under the Seventh Framework Programme on Energy Efficiency, only seven (3%) were related to citizen science. Although there is a large volume of publications on energy efficiency (over 200,000) and a considerable number of publications on citizen science (>30,000 articles), only 336 documents were identified that deal with both topics. The number of projects and publications on these topics has increased in recent years, with universities being the institutions that have published the most. Content analysis found that the most frequent topics are public perception of the use of renewable energies; citizen participation in measures to address climate change and global warming; and the involvement of different stakeholders in the use and responsible consumption of energy. Finally, information was collected on the impact of these publications on social media and altmetric tools. It was revealed that 33% of the 336 papers have had a presence in different sources, especially Twitter. This is a high figure compared with the dissemination achieved by papers from other disciplines.

**Keywords:** energy efficiency; citizen science; scientometrics; altmetric indicators

---

## 1. Introduction

### 1.1. Energy, a Key Sector for Development

Energy generation and consumption are growing in most countries, particularly in the most industrialized countries, and as they grow, they deplete natural resources and damage the environment, posing enormous challenges in a scenario of climate change [1].

Energy efficiency is therefore vital, socio-economically, environmentally, and strategically, in rising to such challenges related to environmental sustainability. It is a key to economic progress, for energy savings delivers economic savings with an impact on the Gross Domestic Product (GDP) and the labour market, with the advent of job-creating businesses (manufacturing, transport, building construction)

engaging in the provision of cross-sectional goods and services and technological innovation [2]. At the same time, indisputable environmental benefits are to be gleaned from responsible natural resource use and CO2 emissions abatement, which help lower the foreign energy dependence that carries a high price tag for many countries. Enhancing energy efficiency therefore calls for investment in energy-producing services [3]. Energy efficiency means acquiring or producing services while minimizing energy consumption and its economic and environmental costs [4].

Given the economic, environmental, and social significance of energy efficiency, the energy industry has been and continues to be crucial for both national governments' and supranational bodies' Research & Development + Innovation (R&D+I) policies, and it has often been the object of legislation. The European Union's leadership in furthering sustainable development dates to the Fifth Ministerial Conference, "Environment for Europe", held in Kiev in 2003 [1], which adopted several protocols on ways to assess environmental impact in cross-border contexts [5]. A few years later, Directive 2009/28 EC of the European Parliament and of the Council set out a series of targets for 2020. On 25 October 2012, Directive 2012/27/EU of the European Parliament and of the Council established a reference framework of measures to further energy efficiency and "ensure the achievement of the Union's 2020 20% headline target on energy efficiency and to pave the way for further energy efficiency improvements beyond that date". In addition, to give continuity to and complete the Millennium Development Goals, the United Nations approved in 2015 the Sustainable Development Goals (SDA) within the Sustainable Development Agenda of 2030. The 17 goals are strategic for achieving a sustainable future for humanity and the planet [6]. Among them, energy has an important role that has been addressed in the SDA 7 (ensuring access to affordable, reliable, sustainable, and modern energy for all).

Growing academic interest in the subject of energy efficiency has led to hosts of research projects conducted under R&D calls such as those included in the European Framework Programmes, along with the publication of any number of scientific papers. In the wide range of topics included in energy efficiency, from the academic point of view, research has been conducted on energy consumption; intelligent energy; renewable energy; environmental recovery; energy management; electrical network; and electric vehicles. This has led to different specific projects on topics such as wind turbines; solar thermal energy; offshore wind farms; energy management; autonomous energy supply devices; electric vehicles; electric motors; oxyfuel combustion; reactor fuels; post-combustion; bioethanol production; and bioenergy carriers [7].

While research on energy efficiency has attracted the attention of researchers for its potential to improve the quality of life of citizens, certain questions might be posed, including the following: Have citizens' needs been considered when designing scientific projects? Have scientific developments had an impact on local communities?

*1.2. Citizen Science, Science Meets Society*

In the different definitions of citizen science (CS) [8], the term is used to describe both a research method and a movement tending to democratize research. Citizen science is also an idea that expresses societal capacity to produce knowledge usable as an evidence-based decision-making tool [9]. Some authors have utilized the expression to define a series of activities that associate citizens with scientific research, narrowing the science-society gap and fostering citizen participation in the research effort [10,11].

In a pioneering citizen science (CS) initiative sponsored by the Audubon Society in 1989, 225 U.S. citizens participated in gathering rainwater samples to determine their acidity [12]. In the mid-nineteen nineties, Irwin [13] described citizen science as "a form of science developed and enacted by citizens themselves". Silvertown [14] later deemed it to be an activity that enabled all citizens to participate in the scientific process, performing tasks traditionally reserved to scientists, from the choice of research subject to the implementation of some of the procedures involved. Today it is defined as active public engagement in science in which citizens participate with their knowledge, tools or resources, while the primary objective is the co-creation of scientific culture and knowledge transfer [15].

According to Mejlgaard et al. [16], this new way to "do science" carries significant social benefits, for it enhances scientific literacy and citizens' critical capacities, democratizes scientific processes, and motivates young people to pursue scientific careers. Researchers themselves also benefit, for the new knowledge generated carries prospects for innovative research. This new scientific ethos has drawn considerable institutional endorsement; the European Union's Framework Programmes, for instance, have funded several citizen science-related projects. In recent years, in addition to gaining visibility in the research community [17,18], citizen science has attracted more and more political, institutional, and public attention. The European Union's framework programmes constitute a good example; they have funded a series of citizen projects, including Foster+, with 13 partner institutions in eight countries, and the Socientize and Citi-sense plans under Horizon 2020 [18]. The concept of citizen science has been added to the European Commission's political agenda as one of the five strategies proposed for the Horizon 2020 "Science with and for Society (SwafS)" programme.

Citizen science therefore constitutes an essential element in digital science and responsible research and innovation (RRI) of a kind that can change the science system. The Socientize Consortium [15] contends that the networking involved in such participation favours the transformation of the scientific system, fostering collective intelligence and collaborative knowledge creation, levelling the research playing field, and encouraging new disciplines and connections to study innovative research questions and topics.

We must emphasise the consideration of the potential of citizen science (CS) alongside traditional research for the implementation of the United Nations Sustainable Development Goals (SDG) [agenda 2030]. In our study, specific attention is paid to those objectives related to energy efficiency and its capacity to improve the quality of life (SDG 7) and other SDGs as SDG 11 (making cities and human settlements inclusive, safe, resilient, and sustainable) and SDG 17 (strengthening the means of implementation and revitalizing the global partnership for sustainable development) [6]

### 1.3. Background and Objectives

Research in each area can be analysed from several perspectives. A few years ago, the predominant model for the production and assessment of academic knowledge revolved around the dissemination of research results as papers and patents. Against that backdrop, scientometrics provided the tools of choice for analysing scientific productivity and studying the impact of research within the academic community [19]. Under the umbrella of scientometrics, bibliometrics is one of the most widely used methodologies, since it enables us to ascertain the dynamics of scientific production in a particular discipline or topic, analyse emerging issues, assess publications' academic impact, and detect the main producers of papers on a given subject (countries, institutions, authors) [20]. Within the field of bibliometrics, some studies focus on fields related to sustainability or alternative energy sources. Scientometric studies have been authored by Romo-Fernández et al. [21] on scientific output in renewable energies, Sanz-Casado et al. [22] on solar energy, Aleixandre-Benavent et al. [23] on scientific production in the area of climate change, Pandiella-Dominique et al. [24] on green and sustainable science and technology dynamics, and Geng et al. [25] on energy consumption and greenhouse gas emissions in the residential sector. Several authors have analysed citation networks in connection with sustainability papers [26] or scientific activity profiles [27,28]. Earlier studies on energy efficiency specifically explored its structure and the possible lines of new research based on scientific papers [29].

Other sources of information, such as research projects, afford a fresh outlook, because they provide insight into ongoing research and new research fronts. Studies have already identified the main areas of interest in European Union framework programmes on energy efficiency [7] and revealed information flows from project formulation through to publication of the findings and their impact in society [2]. More recent studies have begun to use altmetric indicators (alternative metrics based on web 2.0, for analysis of scientific and academic activity [30] to explore the impact of scientific publications beyond the academic world. These indicators were originally conceived as a myriad of

metrics to measure the impact of the broad and varied academic ecosystem [31]. However, it was quickly suggested that they might be capturing "a broader form of impact" [32].

In this way, studies have been conducted in areas such as wind power [33], drawing attention to the interest roused by this new approach. The presence of research in social media and other media enables us to explore a new model of scientific communication based on the academic social web, generating a new area for analysing scientific information [34]. According to Moed [35], the utility of altmetric indicators lies in the immediacy of access after publication of a given paper, the ability to obtain information on its impact on non-academic audiences and the ready analysability of the links between scientific activity and social needs.

In addition to scientometrics and bibliometric studies of publications on sustainability and energy efficiency, studies have also been found that use this methodology to analyse publications on citizen science. Bonney et al. [36] systematically analysed citizen science-related scientific activity as undertaken in different projects and measured its academic impact with quantitative indicators. A study by Comber et al. [37] included a semantic analysis of the terms used to describe citizen science in SCOPUS-listed papers. Follett and Strezov [17] used the Web of Science (WoS) and SCOPUS databases to detect and analyse publications on citizen science and their use in new research projects. Kullenberg and Kasperowski [18] later developed a precise document search strategy to detect the changing use of terms related to WoS-listed publications on citizen science and projects culminating in scientific papers. A more recent case study of citizen participation analysed the variations in output on the subject in SCOPUS to contextualize the findings [38].

Building on the foregoing, the primary aim of this study was to ascertain whether energy efficiency-related publications have adopted a citizen science perspective in either methodology or content. The specific targets included the following.

- To detect the relationships between two scientific fields, energy efficiency and citizen science, through the study of scientific projects and publications on both topics.
- To identify and analyse the social impact of energy efficiency and citizen science publications using altmetric indicators.

This study attempts to answer the following questions: Is there a considerable volume of energy efficiency projects and publications that include the citizen science perspective? What are the main topics in publications on energy efficiency and citizen science? Which journals are publishing papers on these topics? Who are the main producers (countries and institutions)? What is the social impact of publications on energy efficiency and citizen science?

The proposed aims were pursued by applying scientometric and bibliometrics methods to scientific publications in international energy efficiency and citizen science databases, as well as the EU's Seventh Framework Programme projects.

A detailed description of the methodology is given in Section 2, and the findings, in Section 3. In the discussion (Section 4), the results are put into context, while the conclusions are set out in the fifth and last section.

## 2. Materials and Methods

The sources of information for this study included the CORDIS and SCOPUS databases (for information on projects and publications), while Altmetric.com was the tool used to analyse publications' impact on different information sources on the internet (including social media).

**CORDIS** is a European Commission website and public repository with information on the research projects funded by the European Union since 1990. The information can be filtered by call, topic, country, and type of result. The information used for this study was drawn from the Seventh Framework Programme, the main legal and economic instrument for funding research in the European Community for the 2007–2013 period. This program was chosen because it includes projects initiated

in or after 2007 and already completed, and thus it could be expected to make a significant volume of projects and outputs (articles published) recoverable by early 2019.

**SCOPUS** is an international, multi-disciplinary citation database founded by Elsevier in 2004 and widely used in bibliometric studies for the breadth of its journal coverage, its inclusion of citation indicators, and its ability to accommodate tools for analysing, monitoring, and visualizing scientific activity. Together with WoS, it is a major source for analysing quality scientific output, although it lists double the number of social science and humanities journals as the Web of Science, and it includes greater coverage of non-Anglo-Saxon countries. The possibility of performing searches and validations by specific fields such as keywords was of added value to this study. After several tools and strategies were used for document retrieval, it became evident that simultaneous consultation in fields such as title, keywords, and abstract can produce a lot of noise. If the searched-for terms appear in a document's abstract, then that document is retrieved, even if the desired terms are not the main topic. Therefore, treating and validating keywords independently increases the reliability of the information collected, as already tested in previous studies [39]. In this study, all publications retrieved from the SCOPUS database have been analysed (articles, reviews, conference papers) with the search strategy developed. All document types are analysed. In this text, the terms "articles", "papers", "documents", and "publications" are mentioned as synonyms.

**Altmetric.com** was the application used to find the impact of publications on several internet sources such as: fbwalls (number of pages shared on Facebook), feeds (number of blogs mentioning the publication), gplus (number of accounts shared on Google+), msm (number of news sources mentioning the publication), posts (number of different posts that include one or more mentions of the research object (A "post" is any online document that links to one or more research objects (i.e., a post is a mention or a group of mentions)), tweeters (number of Twitter accounts that have tweeted the publication), and videos (number of YouTube/Vimeo channels) [40]. This source, while subject to some limitations (measures, indicators, and platforms are volatile, fleeting and it is difficult to reproduce them when they stop being supported) [41–43], remains one of the most comprehensive tools for studying the visibility of scientific papers beyond the academic arena.

The sources were used to find information on the projects and papers listed below.

(A) Identification of European projects on energy efficiency and citizen science

The CORDIS database was deployed in the following five stages, as set out below.

- Search for energy efficiency projects submitted to the Seventh Framework Programme (data retrieval date: January 2019). Project selection was based on the 65 subjects into which researchers may classify their projects. The projects selected for this study are those on the subject of "energy saving".
- Retrieval of all the information available on each project on the subject of "energy saving" (title, acronym, abstract, applicant, participants, number of participating countries, funding requested, funding granted).
- Search for citizen science-related terms in energy saving projects' abstracts. To identify content on citizen science, keywords were used in the title and abstract of the projects. The keywords were the same as those used to retrieve papers on the subject (see Appendix A).
- Identification of papers resulting from the selected projects. In the SCOPUS database, the project's reference code was searched for in the "Funding Number" field. The publications thus identified were downloaded.

(B) Search strategy to relate energy efficiency papers to citizen science papers in SCOPUS

Publications in SCOPUS on energy efficiency and citizen science were identified through keywords in the texts. From the bibliographic review of both energy efficiency and citizen science, key terms were identified that were successfully used in previous studies [17,26,44]. The resulting search strategy

was applied to the title, abstract, and keywords fields, using the SCOPUS advanced search feature. All manner of documents were retrieved, irrespective of type, language or date of publication. The full strategy used is described in Appendix A.

(C)    Document retrieval in SCOPUS

The six stages involved in document retrieval are listed below.

- Identification of publications on either energy efficiency or citizen science (all document types have been considered)
- Identification of documents on energy efficiency citing citizen science references.
- Identification of documents on citizen science citing energy efficiency references.
- Review of documents using specific keyword searches.
- Retrieval of publications dealing with the selected projects (data retrieval date: January 2019).
- Elimination of duplicate records.

(D)    Bibliometric analysis

The publications gathered from CORDIS and SCOPUS were entered in a MYSQL relational database to obtain the following bibliometric indicators.

- Activity: number of documents per year, compound annual growth rate (CAGR), language, document type, publishing journal, country, and institutional output. For the calculation of the production by countries or institutions, the "total count" is considered: even if a document is signed by different centres or countries, each of them is counted for the whole document. Unlike the "fractional count" (which distributes the publication among the number of signatories), the total count is the system commonly used in bibliometric studies because it does not penalize scientific collaboration [45]
- Specialization: main subject categories on which output is focused. SCOPUS' thematic classification into 27 categories was used, based on the classification of the journals that published the documents.
- Visibility: as is usual in bibliometric studies, the journal's quartile (the journal position in a discipline, based on its impact factor) is considered a criterion for visibility [45]

(E)    Content analysis of SCOPUS publications

The contents of the papers retrieved were analysed in greater depth on the grounds of keyword co-occurrence, and clusters of documents were obtained with VosViewer software [46]. Different tests were carried out using author keywords (the keywords assigned by the author) and index keywords (a set of controlled vocabulary terms that the database assigns to each document). Index keywords were used because they are more standardized and are good descriptors of the papers' content. This analysis revealed the main issues addressed by the papers.

(F)    Publications' impact through altmetric indicators.

Altmetric.com offers a synthetic indicator based on the score a publication receives according to the different sources in which it is mentioned. However, the use of this type of index has been criticized for the lack of transparency and reproducibility [40]. To overcome these limitations, the "Score" provided by Altmetric.con was not used. Direct information was obtained on the number of mentions received by the publications analysed in each source considered by Altmetric.com. As is usual in tools for obtaining altmetric indicators, this required the publication's digital object identifier (DOI) and the deployment of the application programming interface (API). The mentions in different sources were retrieved using a script developed by the Carlos III University of Madrid's Information Metrics Studies Laboratory. This script allows one to obtain the same information that is offered on the Altmetric.com website for each document, but simultaneously for all the publications analysed.

The indicators delivered by Altmetric.com included the following.

- Number and percentage of documents with a DOI.
- Number of documents with mentions in the following sources: Facebook, feeds, msm, posts, Wikipedia, policy documents, and Twitter.
- Maximum number of mentions (data retrieval date: October 2019).
- Total number of mentions in each source.
- Scientific journals most prevalently mentioned in social media.
- Altmetric mentions of publications by subject area.

## 3. Results

The findings for each stage of the process are set out below.

### 3.1. European Projects

The CORDIS database search yielded a total of 265 Seventh Framework Programme projects on energy efficiency, seven of which (2.7%) included citizen science-related issues. The projects' characteristics are summarized in Table 1.

Since 2008, there have been initiatives that included citizen participation in different stages of new knowledge production. The subjects addressed varied, including areas such as water management, household and urban energy efficiency, and environmental resource management. From five to nine countries participated in these projects, two of which were headed by Italy (Table 1).

Sixteen publications from these projects were retrieved from the SCOPUS database: GENeric (six publications), HYDROSYS (four publications), CITYFIED (two publications), INSPIRE-GRID (two publications), IREEN, and CITY-ZEN (one publication each).

**Table 1.** Basic information on energy efficiency projects mentioning citizen science.

| Acronym | Title | Topic | Funding Granted (€) | Start Date | End Date | Coordinating Institution | Country | Relation with Energy Efficiency—Citizen Science |
|---|---|---|---|---|---|---|---|---|
| LENVIS | Localised environmental and health information services for all: user-centric collaborative decision support network for water and air quality management | Water management, environment, and energy | 2,232,223 | 1 September 2008 | 31 January 2012 | Univ. degli studi di Milano Bicocca | Italy | Citizen participation is used to collect data on the environment, energy, and health. Various stakeholders from the Netherlands, Portugal, and Italy participate. |
| GENeric | GENeric European Sustainable Information Space for Environment | Inclusive, low-cost technologies for information and communication (ICT) for environmental management | 8,898,432 | 1 September 2008 | 31 August 2011 | Thales Alenia Space | France | It is proposed to develop a low-cost system for citizens to provide information on their health, energy, and environmental needs and to access data on these topics. |
| HYDROSYS | HYDROSYS: Advanced spatial analysis tools for on-site environmental monitoring and management | Environmental resource management | 3,260,611 | 1 June 2008 | 31 May 2011 | Graz Technical Univ. | Austria | ICTs are developed that are accessible to citizens so that they can develop environmental and energy resource monitoring actions. Applications are offered for mobile phones and tablets for different users. |
| IREEN | ICT Roadmap for Energy-Efficient Neighbourhoods | Household energy efficiency | 998,443 | 1 September 2011 | 30 November 2013 | Manchester City Govt. | UK | Different stakeholders are included in the project to define, assess, and analyse their experience on energy management in homes and cities. |
| INSPIRE-GRID | Improved and eNhanced Stakeholder Participation in Reinforcement of Electricity Grid | Electricity grid and citizens | 2,571,834 | 1 October 2013 | 30 September 2016 | Ricerca Sistema Energetico | Italy | Citizen participation in energy efficiency decision-making processes to improve their perception of energy management. |
| CITY-ZEN | City-zen, a balanced approach to the city of the future | Efficient cities | 25,189,520 | 1 March 2014 | 28 February 2019 | Vlaamse Instelling Voor Technologisch | Belgium | Citizen participation is proposed to solve problems related to energy efficiency in cities. Training and awareness-raising campaigns are established for different stakeholders. |
| CITYFIED | RepliCable and InnovaTive Future Efficient Districts and cities | Intelligent cities | 27,004,955 | 1 April 2014 | 31 March 2019 | Fundación CARTIF | Spain | The aim is to develop smart, energy-efficient cities. Citizens from different countries participate by providing data and validating products and processes developed in the project. |

### 3.2. Document Retrieval

The search strategy yielded a series of documents in each stage that met the established criteria. The documents retrieved are described in Figure 1. The documents on energy efficiency (273.403) and those on citizen science (30.962) were identified and collected from SCOPUS database. Despite the large number of publications on both topics, only 60 documents deal with both subjects. Then, documents with references cited on citizen science (95.092) were identified; in this case 190, are related to energy efficiency. The same procedure was carried out with publications containing references on energy efficiency (332.670); 112 of them are related to citizen science. These three subsets were combined with the 16 publications collected in SCOPUS from the seven projects analysed. The duplicate documents were eliminated, and the final number of publications was 336. All of them were analysed in this paper.

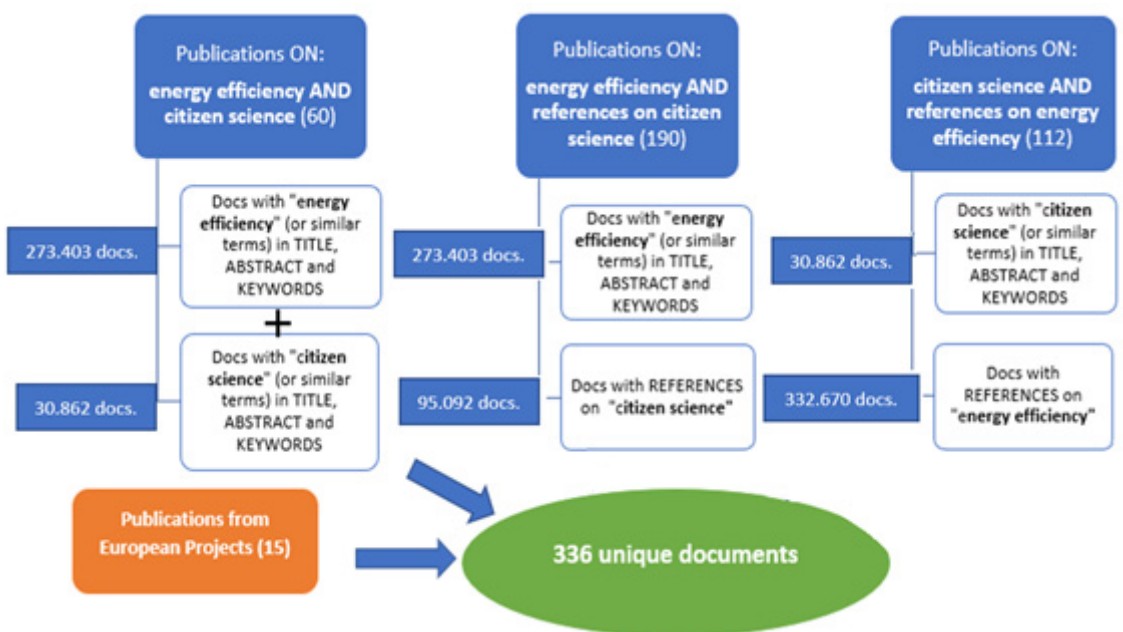

**Figure 1.** Document retrieval process.

### 3.3. Bibliometric Analysis

After deletion of the duplicates, the total number of documents identified, from SCOPUS database, came to 336, with output varying over time as shown in Figure 2. The first document on both energy efficiency and citizen science dated from 1980, while output peaked at 58 publications in 2017. The decline in 2018 was attributed to database updating when the documents were downloaded. In bibliometric studies, it is common to observe that the last year analysed (if very recent) has fewer documents than the rest, due to the late updating of some documents in the database. The Cumulative Average Growth Rate CAGR for the entire period was 9.82%. Given the disparity of data between the earliest and most recent years, however, the sample was divided into equal periods (5 years). Further to that distribution, 92% of the documents were published after 2006, while the largest increases have occurred since 2010.

An analysis of the formal features of the documents showed that English was the predominant language (98.46%), while journal articles, at 68.22%, were the prevalent type, followed by conference papers (16.89%). All the other types accounted for less than 5% of the total.

The dissemination vehicles exhibited considerable scatter, with 234 journals indexed in SCOPUS, counting journals and conferences. The most productive journals (with 15 articles each) were Energy Policy, founded in 1973, and the Journal of Cleaner Production, published since 1993. They were followed by Energy with seven articles and Advanced Material Research, Applied Energy, and Sustainability, with six each. Those six journals hosted more than 16% of the output identified and

exhibited high-quality ratings. From the bibliometric point of view, one of the most used indicators to measure the quality of a journal is the impact factor, which can be used to classify journals into quartiles within each discipline. Analysis of the journals according to the quartile assigned by the Scimago Journal Rank (SJR) shows that four of the journals with the highest number of publications on the target topic lay in the first quartile (Q1). Advanced Materials Research occupied the third or fourth quartile (Q3 or Q4) in the years when it published papers on both subjects, and Sustainability was positioned in the second quartile (Q2). (See Appendix B). This information confirms that energy efficiency and citizen science are not marginal subjects, since they have been well accepted in high-quality journals.

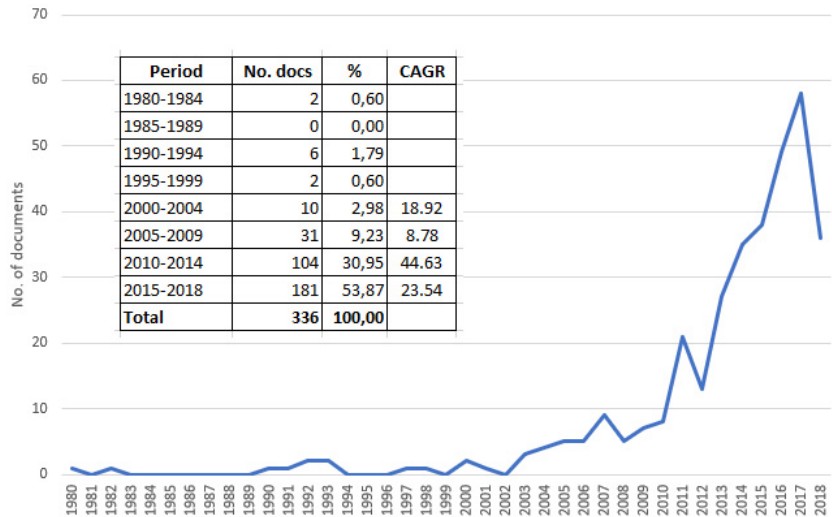

| Period | No. docs | % | CAGR |
|---|---|---|---|
| 1980-1984 | 2 | 0,60 | |
| 1985-1989 | 0 | 0,00 | |
| 1990-1994 | 6 | 1,79 | |
| 1995-1999 | 2 | 0,60 | |
| 2000-2004 | 10 | 2,98 | 18.92 |
| 2005-2009 | 31 | 9,23 | 8.78 |
| 2010-2014 | 104 | 30,95 | 44.63 |
| 2015-2018 | 181 | 53,87 | 23.54 |
| **Total** | **336** | **100,00** | |

**Figure 2.** Number of scientific publications on energy efficiency and citizen science, 1980-2018.

Of the 336 documents analysed, the greatest numbers of publications on energy efficiency and citizen science (135 documents) appeared in journals of environmental science (22%), 108 documents in engineering (18%) and 90 documents on energy (15%) which together accounted for more than half the production. Social sciences ranked fourth, with 85 documents (14% of the publications) (Figure 3).

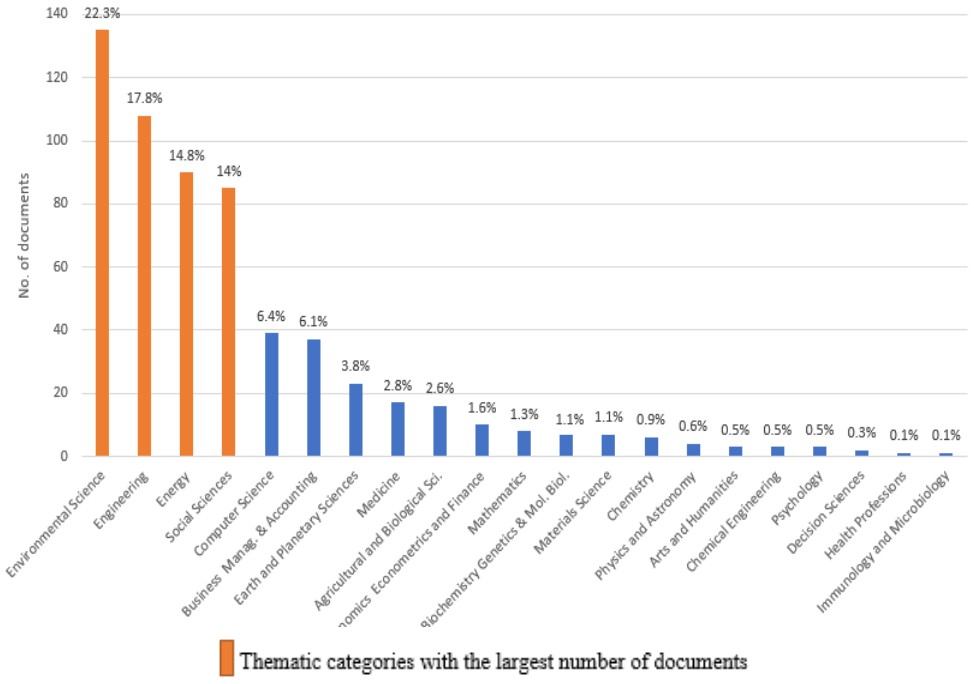

**Figure 3.** Subject area distribution of papers on energy efficiency and citizen science.

The distribution by country was headed by the United States, the United Kingdom, China, Germany, and Australia. Although the papers were signed by a total of 57 countries, only 15 countries published more than five papers in the target period. The most productive institutions (in terms of publications in SCOPUS) were universities, with the University of Sheffield (seven papers) at the top of the list, followed by the Universities of East Anglia and Manchester, Queensland University of Technology, and Cardiff University, with five papers each.

### 3.4. Content Analysis of SCOPUS Publications

In order to ascertain which topics predominate in energy efficiency-citizen science publications, a content analysis based on keyword frequency was performed. "Energy efficiency" was the most cited term, as expected, since it was one of the criteria defined in the initial search strategy. "Energy conservation" and "sustainable development" were also widely repeated. "Public participation" and "local participation" were the two highest frequency expressions in connection with citizen science. Table 2 lists the frequency of keywords present in 15 or more documents on energy efficiency and citizen science. Since all the keywords in the documents are collected, terms not related to energy efficiency or citizen science may appear.

**Table 2.** Most frequent keywords (found in 15 or more documents).

| Keyword | No. of Documents |
| --- | --- |
| Energy efficiency | 147 |
| Energy conservation | 62 |
| Sustainable development | 49 |
| Public participation | 38 |
| Energy utilization | 37 |
| Climate change | 36 |
| Decision making | 34 |
| Energy policy | 32 |
| Sustainability | 28 |
| Housing | 25 |
| Local participation | 22 |
| Economics | 21 |
| Environmental protection | 21 |
| Article | 20 |
| Building | 19 |
| Emission control | 19 |
| China | 18 |
| Human | 18 |
| Public policy | 18 |
| Carbon dioxide | 16 |
| Energy use | 16 |
| United Kingdom | 16 |
| United States | 16 |
| Greenhouse gases | 15 |
| Participatory approach | 15 |

To analyse inter-document relationships, the 336 publications on energy efficiency and citizen science were grouped using clustering analysis based on keyword co-occurrence with VosViewer software [46]. Three predominant clusters were identified: cluster No 1: energy efficiency regulation and construction (red); cluster No. 2: socio-cultural concerns (green); and cluster No. 3: eco-sustainability (blue) (Figure 4). Although these three clusters are the most relevant in terms of number of documents, other smaller ones appear such as the one that groups terms related to citizen science and air quality (violet), another one that contains publications on public engagement and energy management (yellow) and another one with a few documents with terms related to energy transport (turquoise).

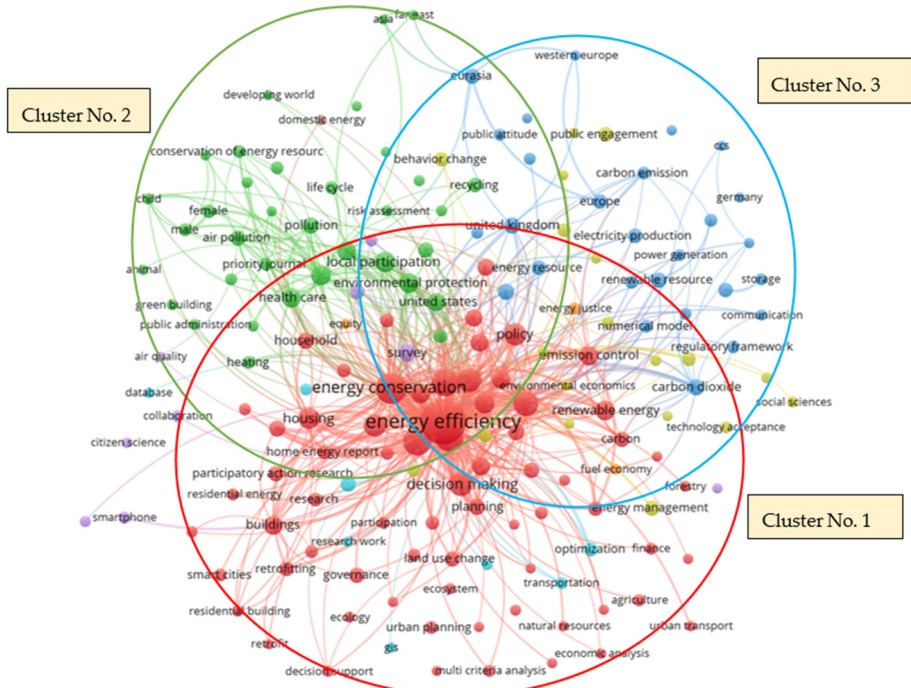

**Figure 4.** Keyword co-occurrence clusters in all the papers dealing with both energy efficiency and citizen science.

## 3.5. Publications' Impact through Altmetric Indicators.

Of the 336 documents retrieved from SCOPUS, 291 (86.61%) had a DOI. An analysis of that set identified 96 mentions in social media and other internet sources (33% of the total number of documents with DOI). As Table 3 shows, all 96 appeared in posts, and most (83.3%) also appeared on Twitter. Interestingly, 17.7% were cited in policy documents. Taken together, the 96 documents were the object of 1551 references in social media and other on-line media.

Seventy (30%) of the 234 journals publishing documents on energy efficiency-citizen science included papers with altmetric indicators, and the 11 journals with two or more papers so cited accounted for 34% of all mentions. Table 4 lists the number of papers with altmetric indicators, grouped by publishing journal and altmetric source. The number of mentions received by the documents is also shown in parentheses. Most of the papers mentioned in on-line media were published in the journals with the highest output. Nonetheless, some of the most productive journals, such as Energy and Advanced Materials Research, were not referenced in social media and other sources. The maximum number of mentions is in post and on Twitter. Likewise, out of the 96 documents with altmetric indicators, these 11 journals concentrate 60% of mentions on Facebook, 70% in feeds, 46% of mentions in post, 25% in msm, 37% in policies, and 46% in Twitter.

**Table 3.** Scientific publications with altmetric indicators.

| Source | No. Docs Mentioned | % Doc | Total Mentions | % Mentions |
|---|---|---|---|---|
| Facebook | 19 | 19.79 | 32 | 2.06 |
| Feeds | 8 | 8.33 | 10 | 0.64 |
| Post | 96 | 100.00 | 864 | 55.71 |
| News (msm) | 9 | 9.38 | 28 | 1.81 |
| Policy docs | 17 | 17.71 | 19 | 1.23 |
| Wikipedia | 3 | 3.13 | 7 | 0.45 |
| Twitter | 80 | 83.33 | 591 | 38.10 |
| **Total** | **96** | **100.00** | **1551** | **100.00** |

**Table 4.** Number of papers with altmetric indicator by journal.

| Journal | No. Docs with Altmetrics Per Journal | Total No. Docs | No. Papers with Mentions (No. of Mentions) | | | | | |
|---|---|---|---|---|---|---|---|---|
| | | | Facebook | Feeds | Posts | msms | Pol. Docs | Twitter |
| Energy Policy | 8 | 15 | 2 (2) | 2 (2) | 8 (91) | 0 | 1 (1) | 8 (63) |
| Building Research & Information | 5 | 5 | 0 | 0 | 5 (58) | 0 | 1 (1) | 5 (28) |
| Energy Research & Social Science | 4 | 5 | 1 (1) | 2 (2) | 4 (76) | 1 (1) | 2 (2) | 3 (46) |
| Applied Energy | 2 | 6 | 0 | 0 | 2 (4) | 0 | 1 (1) | 1 (3) |
| Energy & Buildings | 2 | 4 | 1 (1) | 0 | 1 (7) | 0 | 0 | 1 (5) |
| Environmental Health | 2 | 2 | 0 | 0 | 2 (29) | 0 | 0 | 2 (24) |
| Journal of Cleaner Production | 2 | 15 | 0 | 0 | 1 (4) | 0 | 0 | 1 (4) |
| Renewable Energy: An International Journal | 2 | 2 | 1 (3) | 0 | 2 (28) | 0 | 0 | 2 (21) |
| Sustainability | 2 | 6 | 1 (1) | 0 | 2 (3) | 0 | 0 | 2 (10) |
| Synthesis Lectures on Engineering | 2 | 2 | 0 | 0 | 2 (2) | 0 | 0 | 2 (2) |
| The National Academies Press | 2 | 2 | 1 (11) | 1 (3) | 2 (99) | 1 (6) | 2 (2) | 1 (64) |
| Total | 33 | 64 | 7 (19) | 5(7) | 31 (401) | 2 (7) | 7 (7) | 28 (270) |
| % of total documents with altmetric indicators | 34 | | 37 (60%) | 62 (70%) | 32 (46%) | 22 (25%) | 41 (37%) | 35 (46%) |

In Figure 3, the distribution of the documents on energy efficiency and citizen science according to the thematic classification of the journal in SCOPUS is shown. The subset of 96 documents with altmetric indicators is especially concentrated in environmental science (42.7% of all the documents

with mentions), energy (30.2%) and engineering (25%). As can be seen in Table 5, these first four disciplines show higher percentages of altmetric indicators than the distribution shown in Table 3.

**Table 5.** Number and percentage of documents with mentions in social media by subject area.

| Subject Area | No. Docs. Mentioned | % of Docs. Mentioned Relative to Total |
|---|---|---|
| Environmental science | 41 | 42.71 |
| Energy | 29 | 30.21 |
| Social science | 28 | 29.17 |
| Engineering | 24 | 25.00 |
| Medicine | 17 | 17.71 |
| Earth and planetary sciences | 7 | 7.29 |
| Agricultural and biological sciences | 6 | 6.25 |
| Biochemistry, genetics and molecular biology | 5 | 5.21 |
| Business, management and accounting | 5 | 5.21 |
| Economics, econometrics and finance | 4 | 4.17 |
| Computer science | 3 | 3.13 |
| Chemistry | 2 | 2.08 |
| Arts and humanities | 1 | 1.04 |
| Chemical engineering | 1 | 1.04 |
| Decision sciences | 1 | 1.04 |
| Immunology and microbiology | 1 | 1.04 |
| Material science | 1 | 1.04 |
| Mathematics | 1 | 1.04 |
| Physics and astronomy | 1 | 1.04 |
| Psychology | 1 | 1.04 |
| Health professions | 0 | 0 |
| Total | 96 | 100 |
| Summation | 179 | 186.46 |

## 4. Discussion

Citizen science is acquiring ever greater weight in the present context of globalization and interaction among agents conducting scientific and technological research, as attested to by the steep rise in the number of recent initiatives and further by international and national agencies that fund R&D+I. Europe has in fact set the alignment of scientific endeavour with societal needs as one of its targets in its bid to lead responsible research and innovation. Currently, it is trying to facilitate the relationship between scientists and citizens, in order to enhance collaboration between them to achieve certain scientific objectives, which allow obtaining research and innovation results interesting to citizens and generate benefits for society. The European Commission's commitment to rise to the challenges inherent in the new perspective on inclusive knowledge generation is driving scientific institutions' participation in open science. In that vein, proof of the importance of citizen science lies in its inclusion as one of the eight priorities defined in the Union's "Open Science Policy Platform Recommendations" [47].

Energy efficiency has a strong economic impact and great social interest, which makes it relevant for citizen science. It is also closely related to sustainable development objectives. More specifically,

the universal access to affordable, safe, sustainable, modern energy called for in Agenda 2030 (SDG 7) [5] necessitates not only the generation of new scientific and technological knowledge but also the allocation of a pivotal role of citizen participation in reaching that goal.

Hence the need to further analyse the dynamics of energy efficiency research conducted in the context of citizen science. As noted, the European Union has recognized the importance of citizen science by providing funds for several projects under its framework programs. This has been reflected in the Swafs calls (including citizen science projects on different topics). In the field of sustainability, given the complexity of the research in this field, authors such as Knapp and others [48] propose a methodology employing various collaborative approaches such as translational science, evidence-based practice, knowledge with action, and citizen science. Other researchers see an advantage: Research that includes a citizen science perspective contributes not only to the area's scientific development but also to an increase in the participating citizens' scientific and environmental knowledge and the exercise of transformative social action [49–51]. As an example, we have looked at the experiences reflected in articles on biodiversity, energy efficiency, and sustainable environments from the perspective of citizen science [52–57].

Earlier studies revealed that citizen science projects cover a wide spectrum of areas, including health [58], light pollution [59], astronomy [60,61], and biology/biodiversity monitoring [62]. In general, these are qualitative studies that analyse the dynamics of the projects, the perception of the different stakeholders, and the effects of citizen participation in the different phases of the project. However, no quantitative studies have been identified that analyse the proportion of projects on a specific topic that include a citizen science perspective. Therefore, the novelty of our study is to have quantitative data showing that in the field of energy efficiency, only 3% of the 265 projects under the Seventh Framework Programme on energy saving addressed citizen science-related concerns. These projects focused on topics such as water management, domestic and urban energy efficiency, and environmental resource management, and they included varied methodologies for incorporating citizen opinion into the different phases of the research process. When comparing our results with those of other studies on energy efficiency projects [29], we found that two of the projects analysed here (CITY-ZEN and CITY FIED) were the activities that obtained the most funding under the energy saving topic of the Seventh Framework Programme; they were granted more than 25 million euros. In terms of participation and leadership by country, Germany (which participates in 197 FP7 energy saving projects and leads 44 of them) does not lead any of the projects that include aspects of citizen science. Italy, however, leads two of the projects analysed here, which account for 7% of the total energy saving projects led by that country [27]. Another striking finding is the low number of post-project publications. Other studies on publications on energy saving projects have identified an average of six publications per project [2] (and an even shorter analysis period after the end of the project). Our study revealed that two of the projects finished two months after data collection, so it is possible that they continue producing publications. However, a more realistic explanation for the low number of publications is that companies are largely involved in these projects, so the main results might not be presented to the academic field.

The growing interest in research on energy efficiency and in citizen science has also been evidenced by an increase in the number of scientific publications in the two areas (may also be due to increased funding). In this line, the search strategy employed in this study identified 273,403 documents related to energy efficiency and more than 30,862 documents on citizen science (from 1980 to 2018) in journals indexed in SCOPUS, with a notable concentration in the last five years. Even so, this study identified only 336 papers dealing jointly with energy efficiency and citizen science. This shows how little connection these two areas have had, at least from the point of view of publication production. However, it is encouraging to note that in recent years the number of publications covering both subjects has grown significantly, coinciding with the implementation of the European Union's framework programme projects. In this connection, just as our results show, a review of earlier studies also found that the number of energy efficiency papers grew exponentially between 2000 and 2014 in international

databases such as the Web of Science, which listed 40,000 articles in this period, nearly one quarter of which appeared in the last five years [26]. Papers on citizen science also rose particularly sharply after 2010 in both the Web of Science [17,44] and SCOPUS [38].

In addition to analysing the evolution of the number of publications, this research identified the journals printing the largest number of publications on energy efficiency and citizen science: Energy Policy; Journal of Cleaner Production; Energy; Advanced Material Research; Applied Energy and Sustainability. Other studies analysing the publications on energy efficiency have identified among the journals with the highest production: Optic Express; Journal of Physical Chemistry C; Solar Energy Materials and Solar Cells [2]. These finding show that the first six journals mentioned have some interest in including articles on citizen science.

An analysis of content based on keywords revealed that, among papers on energy efficiency including a citizen science perspective, there is a prevalence of papers on energy conservation and efficiency in buildings and cities [63,64], along with other papers related to sustainable development [65,66]. Another study found that behavioural factors, such as consumers' attempt to save and plan to invest in energy efficiency/solar panels, influence their preference of surcharges and attitudes towards energy efficiency and solar energy [67]; whereas other research analysed participation in actions to tackle climate change and global warming [68]; and different stakeholders participate in responsible energy use and consumption [69].

The main countries for publications on this topic were the United States, the United Kingdom, China, Germany, and Australia, and the most productive institutions were universities, with the University of Sheffield in first position. Similar results have been obtained in previous studies analysing scientific production on sustainability in the Web of Science [70].

In addition to bibliometric analysis, the last phase of our research focused on studying the impact that publications on energy efficiency and citizen science have had on online media and social media. This analysis, based on the use of altmetric indicators, depended heavily on the existence of a DOI. Around 90% of the papers identified here had one, a finding that was consistent with studies reporting wide DOI coverage in many areas [44,71]. Taken together, those facts justified the use of social media impact indicators.

This study also revealed that 33% of the papers with a DOI have altmetric indicators. Although that percentage may seem low, earlier research found the percentage of papers with such mentions range from 15% to 24%, with social science and humanities the areas where the proportions were highest [42]. Twenty-two per cent of energy efficiency papers [2] and up to 66% of studies on citizen science have been reported to have altmetric indicators [44]. A citation rate of 33% in social media for papers dealing with both subjects seems consistent with those values. The number of tweets is one of the most frequent altmetric indicators, a constant in most studies using Altmetric.com [41,42,67].

Thirty per cent of the journals publishing papers on energy efficiency and citizen science were mentioned in social media. No direct relationship between the number of papers published by a given journal and the percentage of citations in social media was found, however.

Another aspect analysed in this paper is the impact of scientific publications beyond the academic field. The growing presence of mentions on sustainability issues in the online media could show the greater interest of researchers to disseminate their research results through these channels, as well as of citizens to access them. Research has already been done on the inclusion of social network analysis, especially looking into impact in newspapers and on Twitter. For example, a study of climate change news in 27 countries shows the increasing coverage of the topic and the wide impact it is acquiring in the press [72]. Other recent research into sustainability news shows that news related to energy efficiency ranks among the most important topics [73].

Regarding the use of Twitter, research done in 2014 pioneers the analysis of the social impact of energy efficiency. The results show that the keyword occurrence frequency of "solar heat" is on a gradually decreasing trend, while "waterpower" and "biomass" maintain a steady frequency of keyword occurrence. More importantly, the occurrence frequency of "wind" tends to sharply

increase after a certain period [74]. Another study along the same lines that uses Twitter as a source of information to find out which topics (related to sustainability) are of most interest to users detects energy efficiency and climate change as two of the leading subjects [75]. Twitter has also been used as a source for the application of sentimental analysis to find out the public's perception of the use of renewable energies [76]. Also, a study concerning the implementation of energy efficiency systems in building construction incorporates content analysis techniques applied to Twitter to find out the public's opinion on the issue and detect the topics of greatest interest to potential users [77]. Similar results were found in our study.

Finally, the combination of tools for the analysis of scientific publications and their impact beyond the academics world is one of the most current trends from a scientometric perspective. In this sense, this paper is related to other recent ones in the field of sustainability that allow us to identify research related to the SDGs and their relationship with energy issues such as those conducted in SDG 7 [78]. All these results allow us to go forward in the development of document analysis methodologies, in order to explore other areas of knowledge and identify the scientific and social impacts of research policies.

## 5. Conclusions

This study has revealed that, although there is a large volume of projects and publications on energy efficiency and citizen science, there are few publications that analyse both topics simultaneously. That may seem unusual, for these subjects might be expected to be clearly inter-related, particularly considering the trend toward gearing research to solve social problems.

The explanation may lie in the constraints on the methodology used; bibliometric analysis is based on research disseminated in journals listed in international databases of repute. That obviates many "invisible practices" such as publication in other vehicles, technology transfer, dissemination, and popularization. In addition, a wide spectrum of activities unrelated to publication (co-creation, citizen assemblies, participatory research) but nonetheless involving active citizen participation may be underway.

In academic circles, the intersection between energy efficiency and citizen science would appear to be a promising area of research, given the obvious consolidation of both scientific fields. At the same time, the willingness of high-quality journals to publish papers on these subjects may encourage such research. This study has detected that 66% of the journals with the highest production for this subject are in the first quartile.

The possibility of mentions of scientific progress in these disciplines in highly visible online, and social media may be another incentive. In this study, 33% of the 336 publications analysed have altmetric indicators, relatively high figures in relation to those revealed in other disciplines. These findings might be relevant because they show that the habits of scientists have been changing towards an increasing use of the academic web and social networks to disseminate their research results. This allows giving more visibility to the research, reaching not only groups of scientists but also various social actors.

Energy efficiency plays a relevant role in the challenges that countries face to achieve Sustainable Development Goals. There is no doubt that a large part of the responses to these challenges must come from the scientific community, but citizen participation is essential for these responses to become realities. In this sense, monitoring scientific production in energy efficiency and knowing its social impact can be of great help to know how these objectives are being met.

Among the future lines of analysis, we propose developing a similar methodology to study the implementation of citizen science methodologies in other areas of research. We intend to explore various scientific fields related to the SDGs to identify in which participatory methodologies are included. Likewise, we will continue to investigate the impact of research with altmetric indicators to determine which topics have the greatest social impact.

**Author Contributions:** Conceptualization, D.D.F.; methodology, D.D.F. and A.P.-D.; software, D.D.F. and A.P.-D.; validation, M.L.L.; formal analysis and investigation, D.D.F., A.P.-D. and M.L.L.; data curation, A.P.-D.; writing—original draft preparation, D.D.F.; writing—review and editing, M.L.L., E.S.-C.; visualization, D.D.F., A.P.-D.; supervision, D.D.F.; project administration, D.D.F.; funding acquisition, E.S.-C. and D.D.F. All authors have read and agreed to the published version of the manuscript.

**Funding:** This project received funding from the European Union's Horizon 2020 Research and Innovation Programme under grant 741657, SciShops.eu. The content of this article does not reflect the official opinion of the European Union. Responsibility for the information and views expressed in the article lies entirely with the authors.

**Acknowledgments:** The methods used were developed under the project entitled "Detection of new research and innovation fronts. Analysis of knowledge flows in the scientific domain, industry and society in the field of energy efficiency" (Spanish Ministry of the Economy and Competitiveness ref. CSO2014-51916-C2-1-R).

**Conflicts of Interest:** "The authors declare no conflict of interest."

## Appendix A. Search Strategy

- **Publications on Energy Efficiency**

  TITLE-ABS-KEY("Energy saving" OR "Energy efficiency" OR "Energy storage")

- **Publications on Citizen Science**

  TITLE-ABS-KEY ("citizen* scienc*" OR "communit* science*" OR "participator* research*" OR "participator* action* research*" OR "communit*-based research*" OR "citizen* research*" OR "science* shop*" OR "citizen* scient*" OR "public-participation" OR "comunity engagement" OR "community-based monitor* OR "crowd science" OR "civic technoscience" OR "community-based auditing" OR "community environmental policing" OR "citizen observatories" OR "participatory science" OR "volunteer monitoring" OR "volunteered geographic information" OR "volun* GIS" OR "street science" OR "locally based monitoring" OR "volunteer-based monitoring" OR "public participation in scientific research" OR "public engagement")

- **Publications citing Energy Efficiency**

  REF "Energy saving" OR "Energy efficiency" OR "Energy storage")

- **Publications citing Citizen Science**

  REF ("citizen* scienc*" OR "communit* science*" OR "participator* research*" OR "participator* action* research*" OR "communit*-based research*" OR "citizen* research*" OR "science* shop*" OR "citizen* scient*" OR "public-participation" OR "comunity engagement" OR "community-based monitor* OR "crowd science" OR "civic technoscience" OR "community-based auditing" OR "community environmental policing" OR "citizen observatories" OR "participatory science" OR "volunteer monitoring" OR "volunteered geographic information" OR "volun* GIS" OR "street science" OR "locally based monitoring" OR "volunteer-based monitoring" OR "public participation in scientific research" OR "public engagement").

## Appendix B.

**Table A1.** Subject category and yearly quartile rating for journals with the highest output on energy efficiency and citizen science (Information is listed from 1999 to 2018, specifying the yearly quartiles as the year the journal was first listed in the database.

| Journal | Subject Category | 1999 | 2000 | 2001 | 2002 | 2003 | 2004 | 2005 | 2006 | 2007 | 2008 | 2009 | 2010 | 2011 | 2012 | 2013 | 2014 | 2015 | 2016 | 2017 | 2018 |
|---|---|---|---|---|---|---|---|---|---|---|---|---|---|---|---|---|---|---|---|---|---|
| Energy Policy | Energy (miscellaneous) | Q1 | Q1 | Q1 | Q1 | Q1 | Q1 | Q1 | Q1 | Q1 | Q1 | Q1 | Q1 | Q1 | Q1 | Q1 | Q1 | Q1 | Q1 | Q1 | Q1 |
| | Management, Monitoring, Policy & Law | Q1 | Q1 | Q1 | Q1 | Q1 | Q1 | Q1 | Q1 | Q1 | Q1 | Q1 | Q1 | Q1 | Q1 | Q1 | Q1 | Q1 | Q1 | Q1 | Q1 |
| Tabla A1.Journal of Cleaner Production | Environmental Science (miscellaneous) | Q2 | Q1 | Q2 | Q1 | Q2 | Q1 | Q2 | Q1 | Q1 | Q1 | Q1 | Q1 | Q1 | Q1 | Q1 | Q1 | Q1 | Q1 | Q1 | Q1 |
| | Industrial & Manufacturing Engineering | Q2 | Q1 | Q1 | Q1 | Q1 | Q1 | Q1 | Q1 | Q1 | Q1 | Q1 | Q1 | Q1 | Q1 | Q1 | Q1 | Q1 | Q1 | Q1 | Q1 |
| | Renewable Energy, Sustainability and the Environment | Q2 | Q1 | Q2 | Q2 | Q2 | Q1 | Q2 | Q1 | Q1 | Q1 | Q1 | Q1 | Q1 | Q1 | Q1 | Q1 | Q1 | Q1 | Q1 | Q1 |
| | Strategy & Management | Q3 | Q2 | Q2 | Q2 | Q2 | Q1 | Q2 | Q1 | Q1 | Q1 | Q1 | Q1 | Q1 | Q1 | Q1 | Q1 | Q1 | Q1 | Q1 | Q1 |
| Energy | Building & Construction | Q1 | Q1 | Q1 | Q1 | Q1 | Q1 | Q1 | Q1 | Q1 | Q1 | Q1 | Q1 | Q1 | Q1 | Q1 | Q1 | Q1 | Q1 | Q1 | Q1 |
| | Civil & Structural Engineering | Q1 | Q1 | Q1 | Q1 | Q1 | Q1 | Q1 | Q1 | Q1 | Q1 | Q1 | Q1 | Q1 | Q1 | Q1 | Q1 | Q1 | Q1 | Q1 | Q1 |
| | Electrical & Electronic Engineering | Q1 | Q1 | Q1 | Q1 | Q1 | Q1 | Q1 | Q1 | Q1 | Q1 | Q1 | Q1 | Q1 | Q1 | Q1 | Q1 | Q1 | Q1 | Q1 | Q1 |
| | Energy (miscellaneous) | Q1 | Q1 | Q1 | Q1 | Q1 | Q1 | Q1 | Q1 | Q1 | Q1 | Q1 | Q1 | Q1 | Q1 | Q1 | Q1 | Q1 | Q1 | Q1 | Q1 |
| | Industrial & Manufacturing Engineering | Q1 | Q1 | Q1 | Q1 | Q1 | Q1 | Q1 | Q1 | Q1 | Q1 | Q1 | Q1 | Q1 | Q1 | Q1 | Q1 | Q1 | Q1 | Q1 | Q1 |
| | Mechanical Engineering | Q1 | Q1 | Q1 | Q1 | Q1 | Q1 | Q1 | Q1 | Q1 | Q1 | Q1 | Q1 | Q1 | Q1 | Q1 | Q1 | Q1 | Q1 | Q1 | Q1 |
| | Pollution | Q2 | Q1 | Q1 | Q1 | Q1 | Q2 | Q1 | Q1 | Q1 | Q1 | Q1 | Q1 | Q1 | Q1 | Q1 | Q1 | Q1 | Q1 | Q1 | Q1 |

**Table A1.** *Cont.*

| Journal | Subject Category | 1999 | 2000 | 2001 | 2002 | 2003 | 2004 | 2005 | 2006 | 2007 | 2008 | 2009 | 2010 | 2011 | 2012 | 2013 | 2014 | 2015 | 2016 | 2017 | 2018 |
|---|---|---|---|---|---|---|---|---|---|---|---|---|---|---|---|---|---|---|---|---|---|
| Advanced Materials Research | Engineering (miscellaneous) | | | | | | | | Q2 | Q3 | Q3 | Q3 | Q3 | Q3 | Q4 | Q4 | Q4 | Q4 | Q4 | Q4 | Q4 |
| Applied Energy | Building & Construction | Q3 | Q2 | Q2 | Q1 | Q1 | Q1 | Q1 | Q1 | Q1 | Q1 | Q1 | Q1 | Q1 | Q1 | Q1 | Q1 | Q1 | Q1 | Q1 | Q1 |
| | Civil & Structural Engineering | Q3 | Q2 | Q2 | Q2 | Q1 | Q1 | Q1 | Q1 | Q1 | Q1 | Q1 | Q1 | Q1 | Q1 | Q1 | Q1 | Q1 | Q1 | Q1 | Q1 |
| | Energy Engineering and Power Technology | Q2 | Q2 | Q1 | Q1 | Q1 | Q1 | Q1 | Q1 | Q1 | Q1 | Q1 | Q1 | Q1 | Q1 | Q1 | Q1 | Q1 | Q1 | Q1 | Q1 |
| | Engineering (miscellaneous) | Q2 | Q1 | Q1 | Q1 | Q1 | Q1 | Q1 | Q1 | Q1 | Q1 | Q1 | Q1 | Q1 | Q1 | Q1 | Q1 | Q1 | Q1 | Q1 | Q1 |
| | Fuel Technology | Q2 | Q1 | Q1 | Q1 | Q1 | Q1 | Q1 | Q1 | Q1 | Q1 | Q1 | Q1 | Q1 | Q1 | Q1 | Q1 | Q1 | Q1 | Q1 | Q1 |
| | Management, Monitoring, Policy & Law | Q3 | Q2 | Q2 | Q1 | Q1 | Q1 | Q1 | Q1 | Q1 | Q1 | Q1 | Q1 | Q1 | Q1 | Q1 | Q1 | Q1 | Q1 | Q1 | Q1 |
| | Mechanical Engineering | Q3 | Q2 | Q2 | Q2 | Q1 | Q1 | Q1 | Q1 | Q1 | Q1 | Q1 | Q1 | Q1 | Q1 | Q1 | Q1 | Q1 | Q1 | Q1 | Q1 |
| | Nuclear Energy & Engineering | Q3 | Q2 | Q2 | Q1 | Q1 | Q1 | Q1 | Q1 | Q1 | Q1 | Q1 | Q1 | Q1 | Q1 | Q1 | Q1 | Q1 | Q1 | Q1 | Q1 |
| Sustainability | Geography, Planning & Development | | | | | | | | | | | | Q3 | Q2 | Q2 | Q2 | Q2 | Q2 | Q2 | Q2 | Q2 |
| | Management, Monitoring, Policy & Law | | | | | | | | | | | | Q4 | Q3 | Q2 | Q2 | Q2 | Q2 | Q2 | Q2 | Q2 |
| | Renewable Energy, Sustainability & the Environment | | | | | | | | | | | | Q4 | Q3 | Q2 | Q2 | Q2 | Q2 | Q2 | Q3 | Q2 |

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
