# Peer review of "Scientometric Analysis of Research in Energy Efficiency and Citizen Science through Projects and Publications"

_sustainability, doi:10.3390/su12125175_

Round 1
Reviewer 1 Report
Comments:
Some of these comments may be easily addressed in the paper, for example, #1, #4, and #5. If comments #2 and #3 are too involved, then I would request that the authors recommend this kind of analysis in future research and as a limitation to this study.
- Table 2 isn't necessary and the information could be summarized in text. The table shows the SJR quartile rankings for the journals, which is fine, but this doesn't indicate any correlation or causation. In the discussion section, the authors explain that this information shows that authors are publishing these types of studies in high traffic journals, and so this seems to be the motivation for including this section. That motivation needs to be articulated within the section, though, and not only in the discussion.
- Table 3 lists the most frequent keywords attached to the articles in Scopus (I think -- could be clearer about the database and whether these were author supplied keywords or assigned in Scopus). This result is interesting but could be more interesting if it was correlated with the results presented in Table 6, especially those keywords and subject areas that both deal with what the articles are about.
- The keyword co-occurence and clustering analysis are the most interesting parts of this manuscript, and it would have been great if this method was applied to some of the other sections (e.g., see comment #2).
- It would be helpful if the authors defined social media terms. For example, what are 'posts'? What platform are these 'posts' from? What is 'msms'?
- Do the authors have any information on whether any of the social media posts involved discussions? Or were they primarily re-tweets and like? More information here would be helpful.
Minor comments:
- Minor grammatical checking needed
- subsection numbers in section 3 should be double checked -- it looks like they go from 3.3 to 4.4 before section 4 begins
Author Response
Response to reviewers
We welcome the reviewers' comments and believe that the inclusion of their suggestions has improved the clarity and the quality of the paper. Afterward, we mention the changes made to the text. These changes are marked with change control in the original document.
Reviewer 1:
Some of these comments may be easily addressed in the paper, for example, #1, #4, and #5. If comments #2 and #3 are too involved, then I would request that the authors recommend this kind of analysis in future research and as a limitation to this study.
- Table 2 isn't necessary and the information could be summarized in text. The table shows the SJR quartile rankings for the journals, which is fine, but this doesn't indicate any correlation or causation. In the discussion section, the authors explain that this information shows that authors are publishing these types of studies in high traffic journals, and so this seems to be the motivation for including this section. That motivation needs to be articulated within the section, though, and not only in the discussion.
Answer
Table 2 was removed from the text and included in the Appendix. A commentary on the quality of the journals has been included in the results section.
- Table 3 lists the most frequent keywords attached to the articles in Scopus (I think -- could be clearer about the database and whether these were author supplied keywords or assigned in Scopus). This result is interesting but could be more interesting if it was correlated with the results presented in Table 6, especially those keywords and subject areas that both deal with what the articles are about.
Answer
In the methodological section the explanation on the use of keywords has been extended
It has not been possible to extend the study to relate the data from the clusters to those in table 6. A comment on this issue has been included in the conclusions.
- The keyword co-occurence and clustering analysis are the most interesting parts of this manuscript, and it would have been great if this method was applied to some of the other sections (e.g., see comment #2).
Answer
We agree with the reviewer's comment but methodologically it has not been possible for now. However, we will continue working in order to elaborate the necessary scripts for the future analysis of these aspects. In the conclusions section we have included some comments on this way.
- It would be helpful if the authors defined social media terms. For example, what are 'posts'? What platform are these 'posts' from? What is 'msms'?
Answer
A more detailed explanation of the sources has been introduced in the methodology section. A reference to the website with information on the sources included in Altmetric.com has been included
- Do the authors have any information on whether any of the social media posts involved discussions? Or were they primarily re-tweets and like? More information here would be helpful.
Answer
Information about the sources used by Altmetric.com has been included and a footnote defining the scope of the posts has been incorporated.
- Minor grammatical checking needed
Answer:
Grammatical corrections have been made. Since the text has been translated by a professional translator, we would appreciate it if the reviewer or editor could let us know if any other specific corrections are necessary.
- subsection numbers in section 3 should be double checked -- it looks like they go from 3.3 to 4.4 before section 4 begins
Answer
Subsection numbers in section 3 has been reviewed and modified

Reviewer 2 Report
Dear Authors,
The idea of your article is quite interesting since it approaches a recent trend: making research more close to the real interests and concerns of the population.
Your approach is quite sound, yet I might suggest trying to compare the topic and themes of the energy related scientific publications with the results generated by google or other search engines concerning people's inquiries regarding sustainable energy, heat pumps, solar and photovoltaic panels, wind turbines and so on.
In the Conclusion part you could refer how you relate to previous research in this area if there are any.
Minor improvements in language are in order.
I recommend accept publishing after minor revisions.
Author Response
Response to reviewers
We welcome the reviewers' comments and believe that the inclusion of their suggestions has improved the clarity and the quality of the paper. Afterward, we mention the changes made to the text. These changes are marked with change control in the original document.
Reviewer 2:
1-The idea of your article is quite interesting since it approaches a recent trend: making research more close to the real interests and concerns of the population.Your approach is quite sound, yet I might suggest trying to compare the topic and themes of the energy related scientific publications with the results generated by google or other search engines concerning people's inquiries regarding sustainable energy, heat pumps, solar and photovoltaic panels, wind turbines and so on. In the Conclusion part you could refer how you relate to previous research in this area if there are any.
Answer:
In the discussion, new bibliography has been included relating our study to research using social networks such as Twitter or press (no relevant studies on Google have been found)
2-Minor improvements in language are in order.
Answer:
Grammatical corrections have been made. A professional translator has translated the text.
We would appreciate it if the reviewers or editor could let us know if any other specific corrections are necessary.

Reviewer 3 Report
Please, see the attached file.

Author Response
"Please see the attachment."

Round 2
Reviewer 3 Report
- General information
The authors significantly improved the quality of the manuscript. Evidently, a lot of work has been put into the data analysis. The presented methodology may be useful for other researchers working on similar problems.
However, the authors failed to mention certain aspects, which may have affected the obtained results (e.g. references in the social media having only a political character, since "Energy Efficiency" is a very trendy topic, or not being strictly related to citizen science).
The last section: Conclusions should repeat the most interesting findings in terms of numbers.
A reference point, i.e. relation between the citizen science and other fields of science, is still missing, as far as numerical values are concerned.
- Detailed remarks:
- line 29: "It was detected that 33% of these papers"; it is in not clear what papers are meant; I presume the said 336;
- line 213: "The five stages involved"; rather: "The six stages"
- line 424: "and leads 44"; I suggest to add: "and leads 44 of them"
- line 443: "the final years"; I suggest to specify the years
- I suggest to exchange "detected" with "revealed" throughout the text
- line 413-414: would you be able to provide this information also in terms of numbers?
- I suggest to add in the Introduction the aspect of "energy efficiency" related to longer time of operation of devices with autonomous power supply.
